

# Tweet success? Scientific communication correlates with increased citations in Ecology and Conservation

Clayton T. Lamb[1], Sophie L. Gilbert[2] and Adam T. Ford[3]

[1] Department of Biological Sciences, University of Alberta, Edmonton, Alberta, Canada
[2] Department of Fish and Wildlife Sciences, University of Idaho, Moscow, ID, United States of America
[3] Department of Biology, University of British Columbia, Kelowna, British Columbia, Canada

## ABSTRACT

Science communication is seen as critical for the disciplines of ecology and conservation, where research products are often used to shape policy and decision making. Scientists are increasing their online media communication, via social media and news. Such media engagement has been thought to influence or predict traditional metrics of scholarship, such as citation rates. Here, we measure the association between citation rates and the Altmetric Attention Score—an indicator of the amount and reach of the attention an article has received—along with other forms of bibliometric performance (year published, journal impact factor, and article type). We found that Attention Score was positively correlated with citation rates. However, in recent years, we detected increasing media exposure did not relate to the equivalent citations as in earlier years; signalling a diminishing return on investment. Citations correlated with journal impact factors up to ∼13, but then plateaued, demonstrating that maximizing citations does not require publishing in the highest-impact journals. We conclude that ecology and conservation researchers can increase exposure of their research through social media engagement and, simultaneously, enhance their performance under traditional measures of scholarly activity.

## INTRODUCTION

Communicating science to policymakers, other scientists, and the public is an increasingly important task in an era of "alternative facts" (*Galetti & Costa-Pereira, 2017*). Scientists are finding new means to communicate science using a wide array of online media (e.g., Twitter, Facebook, blogs; *Piwowar, 2013*; *Bornmann, 2014*; *Donner, 2017*). The shifting nature of modern science communication is particularly relevant to the fields of ecology and conservation (E&C), where science is often used to identify and mitigate pressing environmental problems (*Hoffmann et al., 2010*), and to inform the public about such issues. However, an outstanding question from the efforts to diversify the channels of science communication is the extent to which bibliometrics and social media exposure are linked: how can scientists most effectively invest in social media to promote their research?

Corresponding author
Clayton T. Lamb, ctlamb@ualberta.ca

Publications that receive more attention on social (eg. Twitter, Facebook and blogs) and traditional (eg. news and radio) media reach a more diverse, non-scientist group than a publication with a lower media profile. For example, on Twitter—a platform scientists often use to discuss science amongst one another—up to 40% of followers from E&C scientists may be non-scientists, media, and environmental groups (*Darling et al., 2013*). Recent work by *King, Schneer & White (2017)* demonstrated policy decisions could be shaped through policy-based media, which in turn increased public discussion of policy by ~62.7% on social media, compared to baseline levels. For example, the recent ban on grizzly bear (*Ursus arctos*) hunting in British Columbia, Canada (http://bit.ly/2GCrk1W) highlights a change in policy due to social momentum (*Artelle et al., 2014*; *Darimont, 2017*) generated largely on social-media platforms. Given the importance of social media in communication, there has been a proliferation of research on the value of various alternative metrics of science communication (hereafter "altmetrics") for measuring broader impacts and predicting important bibliometrics such as citation count (e.g., *Thelwall et al., 2013*; *Bornmann, 2014*; *Haustein, 2016*; *Finch, O'Hanlon & Dudley, 2017*).

A key feature of altmetrics is that they accumulate rapidly after article publication and often have effectively stopped accumulating, or accumulate very little, before the paper's first citation (*Eysenbach, 2011*). This sequence occurs because the content of publications becomes public knowledge at or right after the publication date (especially for journals with a media embargo policy, like *Science* and *Nature*), whereas publications citing this work may not be available for years after the original work was published. As such, media exposure—including social media—may either influence or forecast the citation rates of a paper. For example, *Eysenbach (2011)* showed that tweets can predict highly cited articles within three days of publication, and *Finch, O'Hanlon & Dudley (2017)* showed that tweets about ornithology papers predict citation rates in a subset of avian-ecology journals. Consequently, altmetrics present a convenient way to rapidly quantify communication of E&C research and may allow for identification of high-impact papers considerably faster than traditional citation rates, which are slow to accumulate.

While there are many types of new media covered under the umbrella of "altmetrics", it is currently unknown which altmetric types best reflect effective scientific outreach to both the public and scientists, which may vary by discipline (*Haustein, 2016*). Citation count and other related bibliometrics determine professional success at many institutions (*Wade, 1975*), but the correlation between altmetrics and bibliometrics varies by altmetric type (*Thelwall et al., 2013*; *De Winter, 2014*; *Haustein, 2016*; *Peoples et al., 2016*), making it difficult for institutions and researchers to prioritize altmetrics generally and for E&C researchers specifically. Further, some altmetrics are vulnerable to manipulation and commercialization, raising concerns regarding their use for evaluation of research impact (*Bornmann, 2014*; *Haustein, 2016*). Determining a single best altmetric predictor of bibliometric performance will likely remain elusive as the online media landscape evolves and new altmetric types emerge. One potential solution to these related problems is to use a broad suite of altmetrics to calculate a combined Altmetric Attention Score (hereafter Attention Score, http://www.altmetric.com; one of most popular 'altmetrics', and on which we focus here). However, the effectiveness of the Attention Score for predicting research

impact has not been evaluated across E&C, or over time (but see *Finch, O'Hanlon & Dudley (2017)* for a focused look at ornithology), leaving a knowledge gap with implications for the evaluation and dissemination of research.

Researchers are under increasing pressure to publish papers that not only have a high level of impact within the scientific community, but also have broader impacts across the public and policy-making spheres (*Donner, 2017*). This is certainly true in the fields of ecology and conservation biology, in part because of the intensifying biodiversity crisis (*Johnson et al., 2017*). As a result, researchers must make difficult decisions regarding allocation of their time and effort in terms of which journals to target and what outreach efforts to undertake. These decisions not only impact the individual researchers' careers, but also society at large if their research succeeds or fails in reaching intended audiences or changing policy and management of natural resources. Here, we examine correlations between traditional bibliometrics (citation rate, journal impact factor), time since publication, and altmetric exposure. We focus on E&C publications, where we anticipate that the growing interest by E&C researchers in social media may be changing the relationship between citation rates and altmetrics.

## MATERIALS & METHODS

### Data

We gathered citation, Attention Score, and other descriptive data (year published, journal impact factor, and article type) on ecology and conservation (E&C) articles published between 2005–2015. This period reflects an era of sufficient social media engagement by researchers to investigate the relationship between Attention Score and citation rates, while allowing sufficient time for more recent articles to acquire citations. Attention Score data was obtained from Altmetric (https://www.altmetric.com/) under a free academic license. Altmetric makes its' data available to academics upon request, but the database is not publicly accessible on their website. The Altmetric data consists of the Attention Score for each paper as well as the counts of individual media sources that comprise the score. Attention Scores are a composite, weighted index of many media sources (https://help.altmetric.com/support/solutions/articles/6000060969-how-is-the-altmetric-score-calculated-). We focused on the most popular and top-weighted media sources: news, blogs, Facebook, Twitter, and Wikipedia.

Citation data were obtained from Scopus (https://www.scopus.com/) using the search terms "Ecolog*" AND "Conservat*" between 2005–2015. Our focus was on research addressing the conservation of nature and ecosystems. To disambiguate this research from work in physics, art, and other fields using the term "conservation", we used the Boolean operator AND "ecolog*". We recognize that this approach excludes some studies relevant to the conservation of nature and ecosystems , but feel that at $n = 39,442$ papers, our search effectively samples the literature of our focal subject. We obtained journal impact factors using Reuter's 2014 impact factor ratings. We merged Scopus and Altmetric data using a unique identifier of the first 30 character of the article and journal titles and the year. Finally, to ensure our citation metrics were comparable between articles, we removed any

methods-based articles, which are often cited more highly than other articles, and were not the focus of our investigation. We removed these articles using the keywords "method*" or "technique*" to cull articles with these words in their article title or journal title. To control for other factors influencing citation rates of articles, we included in all models the number of years since publication, journal impact factor, and article type (review, research article, or letter).

## Modeling approach

We used boosted regression trees (*Elith, Leathwick & Hastie, 2008*) to investigate the relationship between Attention Score and citation rates. Boosted regression trees (BRT) are an advanced form of a generalised linear model (GLM; *Elith, Leathwick & Hastie, 2008*). BRTs were well suited to our application because they can handle the complex, non-linear relationships we expected to find with these data, provide greater predictive performance and are less plagued by multi-collinearity than GLM's (*Elith, Leathwick & Hastie, 2008*). Unlike GLM's, BRTs do not test null hypotheses but instead effectively quantify and illustrate complex, non-linear relationships, such as those expected here. We fit BRTs using the 'gbm' package (*Ridgeway, 2015*) in R (*R Core Team, 2017*). We analyzed the correlation between Attention Score and citation rates in three periods: (1) an early time period of social media uptake (2005–2009); (2) latter period of social media uptake (2010–2015), and (3) and the combined period of our dataset (2005–2015).

A BRT is fitted to data using three main parameters: (1) learning rate: the contribution of each tree to the model. Smaller learning rates result in relatively more trees required to fit the model, with each tree contributing a relatively small amount to the predictions providing a better fit of the model to the data. In general, a lower learning rate is preferred, such that at least 1,000 trees are generated (*Elith, Leathwick & Hastie, 2008*); (2) tree complexity: the number of nodes or splits allowed in each tree. Trees with more nodes are more complex; (3) bag fraction, which is the percentage of data used to train (those data used to build the model) and test (data used to test predictions that were not involved in model creation) the model for each iteration (new tree).

We tested two commonly used learning rates (4 and 8) and tree complexities (0.001, 0.01) and selected as our top model the model that minimized predictive deviance (*Elith, Leathwick & Hastie, 2008*). We calculated the relative influence of each predictor on resulting citation rates and produced response curves. Relative influence is measured by relative number of times variables included in trees weighted by the square root of improvement to the model, averaged over all trees and the influence of each variable scaled so the sum adds to 100 (*Elith, Leathwick & Hastie, 2008*).

## Model validation

We partitioned our data into training (bag fraction = 70%, those data used to build the model) and testing data (30%, data used to test predictions that were not involved in model creation) for each iteration (new tree). We used the testing data and model predictions to calculate predictive accuracy using the coefficient of determination ($R^2$), which we used to assess the generality of the model to predict responses from data not used to generate the

model. Overfitting is reduced in the BRT by optimizing the learning rate and number of trees as described above, but also by using randomness in partitioning of data. The degree of overfitting can be assessed using the model predictive capacity on testing data. BRTs are generally robust to overfitting (*Elith, Leathwick & Hastie, 2008*).

## RESULTS

We gathered—for the 2005–2015 period—39,442 records resulting from the Ecolog* AND Conservation* search from Scopus, and 5,249,064 Altmetric records across all disciplines. The Reuter's 2014 impact factor ratings consisted of 11,718 journal ratings. Merging the Scopus, Altmetric, and impact factor datasets, we produced a final dataset consisting of 8,322 EC articles in 687 different journals, each of which met the search criteria, had an Attention Score available, and was in a journal with an impact factor available. Most articles published during this time received relatively low Attention Scores (<100, Fig. 1), but a few scores exceeded 900. Attention Scores per article have been increasing over the last 10 years and the composition of media sources making up the Attention Score has been shifting (Fig. 1), primarily towards increased Twitter activity.

Not surprisingly, time alone (years since published) increased citation rates, and letters/notes received fewer citations than traditional articles whereas review papers received more citations than both other article types. Models for both time periods produced good predictive accuracy ($R^2 = 0.66$ for 2005–2009 and $R^2 = 0.56$ for 2010–2015). BRT models for both the early (2005–2009)and late periods (2010–2015) reached minimum deviance at 8 trees and a learning rate of 0.01 and 0.001, respectively. The third BRT model assessed the contribution of individual media sources (Facebook, Twitter, news, blogs and Wikipedia) to resulting citation rates. Model predictive accuracy was high ($R^2 = 0.59$). Minimum deviance was reached at 4 trees and a learning rate of 0.001.

Within E&C, we found discipline-specific differences in research impact. Articles with ''conservat*'' in the article or journal title received slightly larger Attention Scores ($9.4 \pm 0.7 \bar{x} \pm$ SEM]). compared to ''ecology*'' in the article or journal title ($7.6 \pm 0.4$). However, articles with ''conservat*'' in the article or journal title received fewer citations ($31.4 \pm 1.3$) than those with ''ecology*'' in the article or journal title ($36.0 \pm 1.5$).

Citation rates were positively correlated with Attention Scores during the 2005–2009 period, and to a lesser extent during the 2010–2015 period. Journal impact factor and time since published were more important during the later period (Fig. 2). Higher Attention Scores generally correlated with increased citations, but an asymptote was present in both time periods (Fig. 3). The association between Attention Scores and citation rates has attenuated over time (Fig. 3), and maximal gains in citation rates were attained at Attention Scores of 68 during the early period, and 530 in the late period, after which the relationship plateaued in both time periods. In both periods citation rates were maximized in journals having impact factors between 11–14. Finally, across the entire 2005–2015 time period, Attention Scores derived from coverage on Blogs, Wikipedia and Tweets had the largest influence on citation rates, while Facebook posts and news articles had the least influence (Fig. 2).
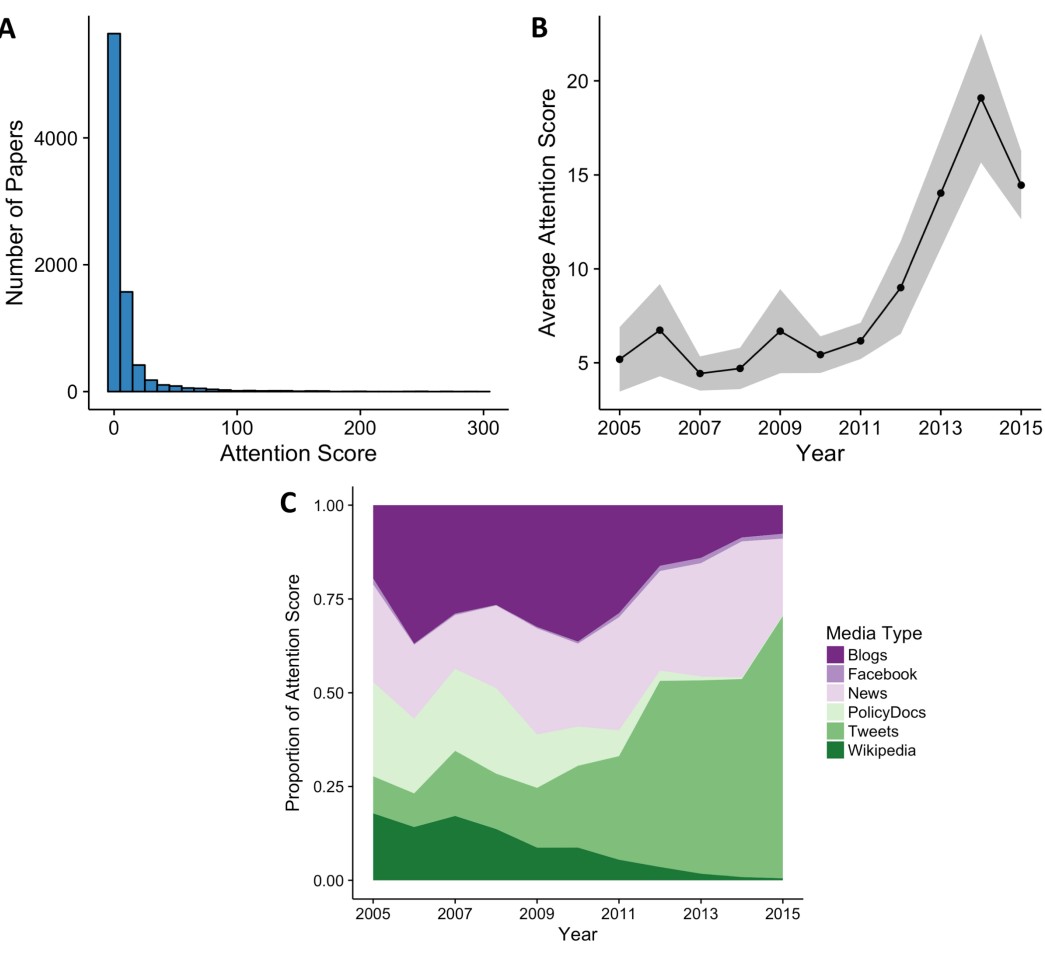

**Figure 1** **Summary stats.** (A) Histogram of Attention Scores for 8,322 ecology and conservation articles published between 2005–2010. Attention Scores were truncated at 300, however, the maximum score for this period was 1,219. 28 articles had Attention Scores exceeding 300. (B) Average Attention Score for ecology and conservation articles between 2005–2015. 95% confidence interval shown in grey. (C) Composition of media sources in Attention Scores between 2005–2015. Starting in 2010, Attention Scores were increasingly composed of tweets from Twitter. By 2015, 70% of the total Attention Score was composed of Tweets.

## DISCUSSION

The fields of ecology and conservation (E&C) have traditionally been linked to applied research, policy, and public engagement (*Lubchenco, 1998*). As such, E&C researchers increasingly rely on social media platforms to promote science to their peers, decision makers, and to the public (*Bickford et al., 2012*; *Darling et al., 2013*; *Priem, 2013*; *Parsons et al., 2014*). Our analyses show that: (1) most published E&C research garners very little attention on social media (e.g., over 80% of articles tracked by Altmertics were tweeted <5 times); (2) social media exposure is positively correlated with citation rates for E&C papers; (3) both journal impact factor and social media exposure on citation show diminishing
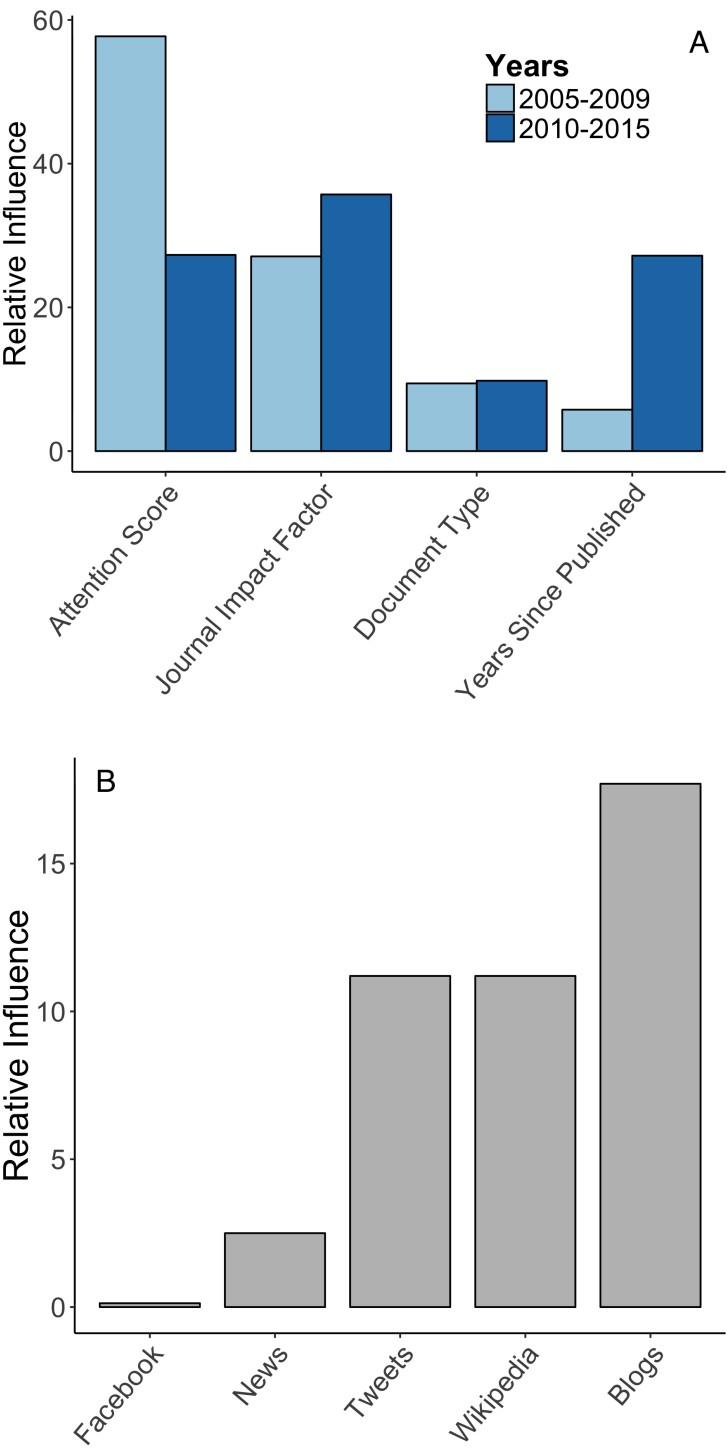

**Figure 2** **Relative influence.** (A) Relative influence of predictive variables, shown for articles published from 2005–2009, and between 2010–2015. Relative influence is measured by relative # of times variables included in trees weighted by the square root of improvement to the model, averaged over all trees (*Elith, Leathwick & Hastie, 2008*). (B) Relative influence of individual media sources on citation rates for the entire period of interest (2005–2015). Policy documents omitted.

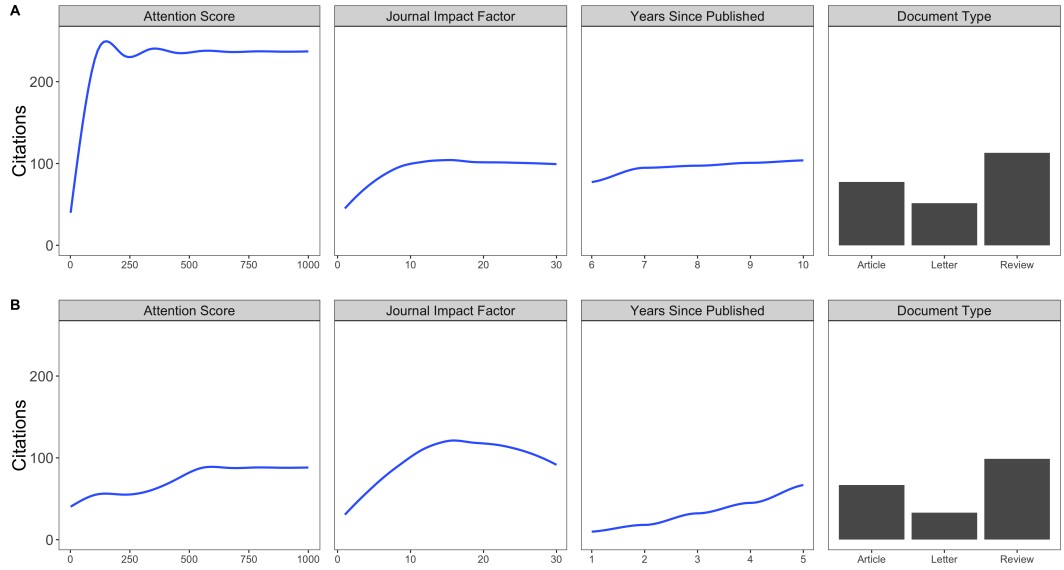

**Figure 3 Response plots.** Response plots showing direction, shape, and magnitude of effects on citation rates. (A) 2005–2009, (B) 2010–2015. We varied each variable from max to min, while fixing the remaining variables at their mean (Attention Score = 10.8, Journal Impact Factor = 4.4, Years Since Published = 5, and Document Type = Article). We quantified the estimated gain in citations per unit increase in Attention Scores during the 2010–2015 period. Assuming 5 years since publication, we estimate the effect of increasing Attention Score on citation rates for three ranges of Attention Scores: low (0–50); moderate (50–540); high (540+). For low Attention Score ranges, every per-unit increase in Attention produces 0.47 citations, requiring about 21 Attention points for each 10-unit increase in citations. For moderate Attention Score ranges, every per-unit increase in Attention Score produces 0.07 citations, such that it takes about 143 Attention points for each 10-unit increase in citations. For high Attention Score ranges, there was no change in citation rates with increasing Attention Score. Attention Scores during the 2005–2009 period were associated with up to four times more citations than the same Score during the more recent 2010–2015 period.

returns in recent years. Below, we discuss the implications of these findings and highlight how E&C researchers can use social media to measure research impact.

The distribution of Altmteric Scores was highly right-skewed, indicating that a few papers can have very wide-reaching attention but most do not. However, average Attention Scores have increased rapidly since 2011—a trend explained, in part, by broader engagement of the public with all forms of online content and the increased use of Twitter to disseminate research (Fig. 1). In addition to this broader societal trend, many researchers are heeding calls to engage in outreach through social media (*Milkman & Berger, 2014*; *Parsons et al., 2014*; *Cooke et al., 2017*). Postdoctoral fellowship programs in E&C, such as the Liber Ero Fellowship (Canada: http://www.liberero.ca), the Smith Fellows (USA: https://conbio.org/mini-sites/smith-fellows), Wilburforce Fellows (USA/Canada: http://www.wilburforce.org/grants/fellowship/), and others provide specialized training in social media engagement for E&C researchers. In the future, graduate and undergraduate E&C students may routinely receive training in social media as part of their studies.

The association between Attention Scores and citation rates varies by type of media within the Attention Score: tweets contributed most to Attention Score for E&C papers,

but blogs had the greatest influence on citation rates. This may signal that researchers turn to blogs as a form of information curation, or that other forms of media (e.g., facebook, twitter, radio, news) create the initial article "hype" which then signals bloggers to cover the article. We cannot discern the causes of these patterns from our analyses, but indeed we observed the latter pattern in a recently published article (*Lamb et al., 2018*) where twitter and news "hype" rapidly accumulated after publication, followed by blogs once the Attention Score had surpassed 200. Finally, the degree and type of social media attention an article receives is also dictated by the article's content, with articles featuring charismatic animals, climate change, or sharing positive news draw more attention (*Papworth et al., 2015*), but we were unable to include these factors in our analysis.

Twitter is a rapidly growing science-communication tool in E&C and likely contributes to the increasing Attention Scores received by articles (Figs. 1B and 1C). Previous work by *Peoples et al. (2016)* found a weak and highly variable relationship between tweet volume and citation rates, whereas we find a stronger, more positive relationship (Fig. S1), likely due to our application of BRT models to address interactive and non-linear effects (*Elith, Leathwick & Hastie, 2008*). Finally, we detected asymptotic relationships between citation rate and each of the media sources comprising the Attention Score. Similar to our study, *Finch, O'Hanlon & Dudley (2017)* also found asymptotic relationships between citation rates, impact factor and Almetric Scores for research focused on the E&C subdiscipline of ornithology. Thus, investigators will likely realize the greatest citation return on investment by diversifying their media outreach channels among blogs, traditional media, twitter, and other outlets for E&C-related subdisciplines.

In spite of the growth in social media activity by researchers, there are asymptotic benefits for traditional measures of scholarly impact (i.e., citation rates). If we assume that social media exposure predicts or contributes towards citation rates (see *Eysenbach, 2011*; *Finch, O'Hanlon & Dudley, 2017*), then our results suggest a diminishing return on investment: it now takes up to four times the Attention Score to achieve an equivalent citation rate as it did 5–10 years ago. This weakening return on investment is consistent with the idea that media consumption is finite (*Rodriguez, Gummadi & Schölkopf, 2014*), and that increasing the number of communicators in a social media network may not increase the amount of media consumed (*Kaplan & Haenlein, 2010*; *Milkman & Berger, 2014*; *Ferrara & Yang, 2015*). This asymptotic relationship between social media and citation rates has important implications for how researchers and institutions should devise media outreach plans, and if/how social media impact can be used to measure research impact. Next steps will be for science communication professionals to work with their research and outreach personnel to optimize strategies.

While our results suggest that an increasing amount of social media attention is needed to generate maximal gains in E&C article citation rates, our results also show that minor increases in social media attention are associated with a steep rise in citation rates for articles with few citations—social media can transform the highly obscure to the notable. This transformation is important, because research impact at many institutions is evaluated both by publication in high impact journals (i.e., impact factors >10) and citation rates— which are positively correlated (Fig. 3, *Wade, 1975*; *Judge et al., 2007*). Since space in

high impact journals is highly competitive, social media can help level the playing field between the few papers accepted into such high-profile outlets and the many more that are rejected. Indeed, we were surprised to discover that the influence of social media exposure on E&C article citation rates was actually far greater than journal impact factor between 2005–2009, and comparable more recently (see Fig. 2). We also found that journal impact factor had diminishing returns on E&C article citation rates, peaking around 13 before levelling off. Combined, these results suggest that, generally, evaluation of research impact should consider discipline-specific asymptotes in media attention and impact factor (i.e., "twimpact factor"; *sensu Eysenbach, 2011*). Finally, our results also suggest that conservationists concerned about reaching a broad audience can do so as effectively with high impact and moderate-impact journals, as has been suggested elsewhere (*Peoples et al., 2016*).

For many E&C researchers, the benefits of social media outreach extend well beyond boosting citation rates. Social media is also a tool to engage with peers, the public, and policy makers from around the world (*Kaplan & Haenlein, 2010*; *Parsons et al., 2014*; *Bombaci et al., 2016*; *Cooke et al., 2017*). Quantifying causal links between research innovation, Attention Scores, citations rates, and policy changes is challenging (e.g., *Danaher, 2017*); yet such linkages are likely why many in E&C fields use social media (*Bombaci et al., 2016*; *Peoples et al., 2016*). Our analysis provides guidance on the potential benefits of social media engagement for research impact. However, we have not identified a specific mechanism linking citations to Attention Scores. A number of factors constrain the effectiveness of science communication in general, including via social media (e.g., the appearance and race of the scientist; *Milkman & Berger, 2014*; *Gheorghiu, Callan & Skylark, 2017*). Moreover, linkages between social media, policy/management change, and public engagement were beyond the scope of our work, but are important avenues of continued inquiry in contemporary scientific communication (*King, Schneer & White, 2017*).

Researchers need to weigh the benefits of social media—potentially enhanced citation rates and public engagement—against the costs of time and risk of exposure (*Cooke et al., 2017*). Understanding how to better harness the power of social media will be a growth area for applied disciplines like E&C, and for evaluation of research impact in the modern era of science communication.

## CONCLUSIONS

Our correlative analysis shows a strong association between science communication (measured by the Altmetric Attention Score) and citation rates. Most online science communication happens within weeks of publication while traditional citations generally begin accumulating months and years later. Pairing the chronology of metric accumulation and an assumption that not all researchers are able to stay up to date with all publications, we believe it is reasonable to suggest that science communication and increasing the profile of one's work may increase citation rates. Of course, to verify this, an experimental approach or additional data to what we had here would be required. We encourage E&C researchers to engage in science communication due to potential benefits such as increased citation rates, networking and public engagement.

### Funding

This work was supported by a Vanier Canada Graduate Scholarship. The funders had no role in study design, data collection and analysis, decision to publish, or preparation of the manuscript.

### Grant Disclosures

The following grant information was disclosed by the authors:
Vanier Canada Graduate Scholarship.

### Competing Interests

The authors declare there are no competing interests.

### Author Contributions

- Clayton T. Lamb conceived and designed the experiments, analyzed the data, contributed reagents/materials/analysis tools, prepared figures and/or tables, authored or reviewed drafts of the paper, approved the final draft.
- Sophie L. Gilbert and Adam T. Ford conceived and designed the experiments, authored or reviewed drafts of the paper, approved the final draft.

### Data Availability

The R script and associated data for analysis are provided as Supplemental Files and the full R script for analysis can be found here: Lamb CT. (2018). Science_Communication_PeerJ. Available at http://doi.org/10.17605/OSF.IO/BUJYR.

### Supplemental Information

Supplemental information for this article can be found online at http://dx.doi.org/10.7717/peerj.4564#supplemental-information.

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
