# Peer review of "Tweet success? Scientific communication correlates with increased citations in Ecology and Conservation"

_PeerJ, doi:10.7717/peerj.4564_

## Round 0.1 · original submission · Minor Revisions

Overall this was a well-written paper on a timely topic of interest to many academics and researchers involved in science communication and outreach. There is, of course, also an important element of communal self-interest in studies like this given the attention paid to citation rates in promotion and tenure decision making. This work therefore provides useful insights for academics involved in such processes.

Both the reviewers agreed that the paper was suitable for publication with minor corrections. In addition to a number of specific corrections noted in the reviews, I would draw attention to three more general issues raised by the reviewers:

1) The need for a stronger explanation of why the authors chose to focus on papers from the discipline of ecology and conservation
2) An explanation of any procedures used to subset the papers returned by their initial search. When I ran the same initial search as the authors on Scopus I returned just under 67,000 papers which is considerably more than you reported in your paper. Without reporting the full syntax and methods of their search it was thereafter difficult to duplicate your results.
3) Reviewer 2 proposes some important interpretations of the plateau effect you note in citations, you will need to clearly address this

In addition I would like to see the Methods clarified a little more. My understanding was that you:

1) Searched for relevant papers within Altmetric
2) Searched for relevant papers with Scopus
3) Married the two sets of search results together using the unique identifier you created.
However, I was unable to follow your precise process for generating papers and associated metrics. You first report the process of getting Altmetric scores and suggest these were obtained via their website. From what I can see they provide a number of tools but no functionality to search for papers from their site - you need to more precisely explain what you used and report the structure of any literature search. It would also be important to note how many, if any, results were returned within one database but not the other (for instance it's my understanding that not all journals are subscribed to Altmetrics).

A number of more technical points can be found on the annotated manuscript attached, please ensure these are addressed fully.

Reviewer 1 ·

Basic reporting

Paper is generally well-written, references provided seems adequate.

Experimental design

It is still not very clear on why the authors have chosen to focus on ecology and conservation (noting that other papers have attempted similar studies linking altmetrics and traditional citation means). Are they doing this just because this hasn’t been explored in the literature? (I hope not!) I’d like to see a stronger framing of the paper’s research focus – what are the questions and why is it necessary for us to find answers for them, and particular so in the context of ecology of conservation?

Additionally, it would be useful if the authors provided more details of their search methodology, given that a search with such generic terms like "ecolog" and "conservat" will end up with a huge number of irrelevant papers (and quite a lot of systematic work would be needed to eliminate these), notwithstanding that effort have been made to cull review papers and letters from the core analysis.

Validity of the findings

Statistical framework seems sound, and is clearly explained.

Additional comments

Social media is playing an increasingly important role in the communication of research findings, especially in the disciplines of ecology and evolution, and biodiversity conservation. As a researcher working in these fields, I fully appreciate the need to regularly use social media platforms to convey interesting findings in digestible snippets, in the hope that some of these information would eventually be utilised by decision makers and policymakers. This paper explore and assesses the relationships between more traditional citation metrics in relation to altmetrics (and counts of media sources) in the context of ecology and environmental research. The authors found that altmetrics were able to predict citation rates until a particular threshold in two time periods. Broadly, the paper is well written, and offers some new insights on how the real-world influence of ecology/conservation research papers could be measured, long before it accumulates citations. The authors have also evaluated a large amount of material before proceeding with their analysis, and their statistical approach seems very sound. I would like to offer a few minor suggestions to frame the paper more clearly (to distinguish it from other papers doing similar things), and strengthen its discussion.

Minor comments
L63 – The (applied) role of ecology/conservation research could be made clearer here, perhaps an example on links between conservation research and policymaking, and a citation on what kinds of environmental problems
L75 – Are there actual instances where policy making processes were shaped from increased lobbying resulting from social media exposure of an environmental/conservation issue
L87 – “showed that tweets” – good to maintain consistence tense
L88 – “ornithology” – please correct this and other spelling mistakes throughout the paper
L90 – Not clearly explained here why is there is need to identify high-impact papers rapidly
L92 – “covered under the umbrella of…”
L95 – I think a more recent citation is needed, since you say it “continues to determine professional success”
L108-111 – Authors say what the paper is going to do, but did not make clear why this is of interest. This section of the paper needs more work to show it is not just another data exploratory paper.
L114 – What kind of descriptive data on these articles?
L123 – Suggest highlighting here that the focus of the paper is on media discourse in the English language – Note that large amounts of similar scientific discourses pass through social media in other language medium (e.g. weibo, weixin)
L124 – More information needed on how the papers were selected after the initial searches – how did you ensured that paper covered content specifically relevant to ecology and conservation? Checking of the abstracts? Checking with the key words of papers? Both disciplines (and search terms) are quite broad and you would likely have had to throw away a lot of non-relevant papers
L133 – you mean “research article”
L136 – “the generalised linear model”
L140 – “GLMs” – please remove the apostrophes throughout – this and for “BRT’s”
L146 – “fitted”
L151 – Rephrase sentence, awkward phrasing
L152 – “percentage”
L188 – please correct spelling of “altmetric” throughout the article
L204 – Not wrong to say this, but conservation is certainly more “applied” compared to ecology, where a lot of research output remains as theoretical or empirical studies
L250 – Perhaps provide an example to show how media outreach by institutions may be revised in response to these patterns
L256 – Could there be other factors involved – that drives up citation rates (e.g. search results algorithms)
L256 – “can transform”
L264 – L269 – This section could be improved by revisiting the findings more in relation the study’s focus on ecology and conservation, the current discussion is a bit generic.

Reviewer 2 ·

Basic reporting

This is a well thought out paper with a specific goal achieved through a correlational analysis of data extracted from two databases – one traditional citation based and the other altimetric based. The authors found relationships between altimetrics and citations.

Which suggest that publicity received by papers has an impact on future citations.
However, there are a few issues that need to be addressed.

- There is a need to clearly define terms as and when they first appear. This would greatly help the reader later in the text. A clear example is the need to clearly define altimetry at the start.

- The study is actually of a specific subset of scientific articles that have both the keywords Ecology AND Conservation. The author should consider changing EC throughout the text to E&C. As the data extracted from the databases were using this operator. Which also means that the study is of a specific smaller subset of a larger field of study. This specific subset of data provides great power for the analysis that the authors have carried out. But it does at the same time only allow inferences for papers written in other parts of the larger field of study. This needs to be acknowledged at the start and in the discussion.

- In Figure 1C, the authors illustrate an important finding of the changing contributing proportion of different publicity channels over time. This is quite an interesting finding worth elaborating on.

- While not a specific issue, could the plateauing of citations with impact factors above 10 be an artefact of the E&C journals themselves? Consider that E&C journals have impact factors below 10 and only a few top tier journals that periodically publish E&C papers (e.g. Nature, Science, PNAS) have impact factors above 10.
The study warrants publication subject to addressing the relatively minor issues highlighted in this review.

Experimental design

The study is well designed to address the relevant question.

More details in the attached PDF.

Validity of the findings

Line 173:
The number of articles found appears to be very low. It could also be because the authors are using AND instead of OR. This means the study is really looking at only a very specific subset of research that include both the keywords Ecolog* and Conservat*.

Line 187-190:
How were discipline-specific (i.e. conservation vs ecology) differences found if the original data search was using the AND operator?  

Line 224-240:
One of the key issues not dealt with in the discussion is that the scope of this study did not include the content of the specific E&C articles. Previous work by Papworth et al and Wijedasa et al. (Papworth et al., 2015; Wijedasa, Aziz, Campos-Arceiz, & Clements, 2013) found that the specific content of the articles were more likely to get featured. While not part of this study, it is a limitation.

Line 263-265:
The levelling off that you noted could also be because there are very few journals in E&C with impact factors greater than 10. So this result may be an artefact of the fact that many journals in E&C have impact factors lower than 10. This is also in contrast to what Papworth et al (2015) found, where the likelihood of being featured in news or twitter was higher for high impact factor journals (e.g. Nature, Science, PNAS).

Additional comments

Well written paper. Most comments made are minor.
Do consider tightening up your results as conclusions drawn on the results as the study is of a specific subset of papers in the field (i.e. those that have both keywords Ecolog* AND Conservat*). This gives considerable power to the study as it may reduce noise.

Annotated reviews are not available for download in order to protect the identity of reviewers who chose to remain anonymous.
External reviews were received for this submission. These reviews were used by the Editor when they made their decision, and can be downloaded below.

---

## Round 0.2 · accepted · Accept

Many thanks for your effort to revise and resubmit this paper, I am happy to accept it for publication. However, please attend to the following issues:

1) Please proof-read the final version carefully, I spotted a few typos, for example:
Line 208 change to "Attention Score on"
Line 259 change to "article's content"

2) Two questions you might like to consider:
I wonder to what extent the declining association between Attention Score and citation rate is a function of the changing profile of the Attention Score itself - for instance policy documents seemed to formerly be an important component but are now barely contributing. Could it be that the change is a result of the fact that scientists differentially select papers to read from different sources - i.e. those in policy documents are seen as "important" but that effect is no longer being captured?

How would you recommend further research should account for factors such as charismatic megafauna, etc. if you believe these variables are important?

External reviews were received for this submission. These reviews were used by the Editor when they made their decision, and can be downloaded below.